# High resolution ensemble description of metamorphic and intrinsically disordered proteins using an efficient hybrid parallel tempering scheme

Rajeswari Appadurai [1], Jayashree Nagesh[2] & Anand Srivastava [1]✉

Mapping free energy landscapes of complex multi-funneled metamorphic proteins and weakly-funneled intrinsically disordered proteins (IDPs) remains challenging. While rare-event sampling molecular dynamics simulations can be useful, they often need to either impose restraints or reweigh the generated data to match experiments. Here, we present a parallel-tempering method that takes advantage of accelerated water dynamics and allows efficient and accurate conformational sampling across a wide variety of proteins. We demonstrate the improved sampling efficiency by benchmarking against standard model systems such as alanine di-peptide, TRP-cage and β-hairpin. The method successfully scales to large metamorphic proteins such as RFA-H and to highly disordered IDPs such as Histatin-5. Across the diverse proteins, the calculated ensemble averages match well with the NMR, SAXS and other biophysical experiments without the need to reweigh. By allowing accurate sampling across different landscapes, the method opens doors for sampling free energy landscape of complex uncharted proteins.

[1] Molecular Biophysics Unit, Indian Institute of Science, Bangalore, Karnataka, India. [2] Solid State & Structural Chemistry Unit, Indian Institute of Science, Bangalore, Karnataka, India. ✉email: anand@iisc.ac.in

**B**iomolecules are not static but exhibit time-dependent dynamic motions that are tightly coupled with their functions. Molecular simulations are now increasingly relied upon for visualizing detailed events of such molecular motions at atomic resolutions, which are often not possible using experimental methods. Nonetheless, sampling in classical molecular dynamics (MD) simulations is limited to the local minima of the free energy landscape. Consequently, accessing the complex energy landscapes of proteins is far from trivial for proteins with multi-step folding[1,2], metamorphic proteins[3–5], and intrinsically disordered proteins (IDPs)[6–8].

Enhanced molecular dynamics simulations such as temperature replica exchange molecular dynamics (TREM)[9], are particularly useful in sampling conformations across energy barriers and therefore expediting the observations of rare biomolecular transitions. In the replica exchange method, conformations are sampled using multiple replicas simulated at a series of low and high temperatures and are stochastically exchanged at regular intervals to provide an unbiased Boltzmann-weighted ensemble of conformations. However, the number of replicas required for the simulation scales exponentially with the degrees of freedom of the system hampering its application in large biomolecular solutions. Bruce Berne and coworkers developed a powerful alternative called replica exchange with solute tempering (REST)[10,11], where they designed the Hamiltonian to partially heat a subset of the system (solute). The approach reduced the required number of replicas by multiple folds. Such a powerful design of the Hamiltonian has been successfully shown to simulate weak binding of Aβ peptide on a lipid bilayer[12], lateral equilibration of lipids in a bilayer[13], and for the generation of conformational ensemble in SH4 unique domain protein[14]. While a much wider application in protein folding is envisioned, however, the method underperforms in proteins where the conformations are separated by large free energy barriers producing poor mixing of replicas between the high temperature regime and low temperature regime. This is likely due to the lack of energy compensation of hot solute and cold solvent. Inclusion of a subset of random water molecules to the central group being tempered improves the mixing but at the expense of poor scaling with the system size[15]. More recently, a generalized version of the REST (gREST) that scales the Hamiltonian based on particles, as well as energy terms[16] further reduces the number of replicas and alleviates the scalability issues to a large extent. However, as the extent of tempering is limited to few energy terms, the method requires longer time to converge on the conformational sampling. This suggests that crossing of the free energy barrier is still a bottleneck for this class of methods.

In general, parallel tempering methods suffer when the barriers between folded/unfolded states and intermediate states are high or when the transition state is diffusive in nature with large entropic barriers slowing down conformational exchanges[17]. The unrealistically reduced residence time in the metastable states prevents switches between conformational basins. In these cases, friction between the polypeptide chain and solvent modulates the speed of protein folding or unfolding[18]. In this work, we propose a hybrid replica exchange method (termed hereafter as Replica exchange with hybrid tempering (REHT)) that differentially and optimally heats up both the solute and solvent. In other words, along with the Hamiltonian scaling of each replica that effectively heats up the protein solute, the replicas are also coupled to different high temperature baths that heat up the system, including solvent particles. The chosen bath temperature range of the replicas is small enough such that the energy difference due to solvent self-interaction is minimal and does not lead to the scalability issue. At the same time, the optimal tempering of the solvent along with the solute ensures an efficient rewiring of the hydration shell that works in cohort with that of protein

conformational change, thereby helping in crossing larger barriers and in particular the entropic barriers. We demonstrate this by applying the protocol on a diverse set of proteins that differ widely in their size and complexity of the underlying free energy landscape. The choice of protein molecules ranges from simple model systems such as alanine dipeptide to fast folders such as TRP-Cage and β-hairpin and proteins with complex energy landscapes such as IDPs (example: Histatin-5) and metamorphic proteins (example: RFA-H).

## Results

We explore the multidimensional free energy landscapes of diversely complex proteins using the REHT method and compare its efficiency with that of the state-of-the-art REST2[11] simulations. Toward this, we exploited the HREX module of PLUMED, originally developed for performing the Hamiltonian replica exchange simulations[19]. The module is very flexible and allows for simultaneous use of different bias in the replicas such as the Hamiltonian, collective variable, temperature and pressure. For REHT method, we include the additional temperature bias in the replicas along with the Hamiltonian scaling of the protein solute and for which we derive and apply the corresponding detailed balance exchange criteria. The detailed derivation of the exchange criteria, honoring detailed balance, is provided in the Methods section. In short, the exchange criteria for REHT is given by:

$$
\Delta_{nm}(\text{REHT}) = - \left[ \begin{array}{c} (\beta_n \lambda_n - \beta_m \lambda_m)\left[H_{pp}(X_n) - H_{pp}(X_m)\right] \\ + (\beta_n \sqrt{\lambda_n} - \beta_m \sqrt{\lambda_m})\left[H_{pw}(X_n) - H_{pw}(X_m)\right] \\ + (\beta_n - \beta_m)[H_{ww}(X_n) - H_{ww}(X_m)] \end{array} \right] \quad (1)
$$

where, $H_{pp}\left(X_{m|n}\right)$, $H_{pw}\left(X_{m|n}\right)$ and $H_{ww}\left(X_{m|n}\right)$ indicates the intra-protein, protein–water and water–water interaction energies in $m^{\text{th}}$ and $n^{\text{th}}$ replicas. $\beta_{m|n}$ and $\lambda_{m|n}$ are the corresponding inverse temperatures and Hamiltonian scaling factor of the two replicas.

Though the methodological advancement of REHT has its origin in REST2, the REHT method has significant impact on the sampling efficiency as will be demonstrated here in this section. The efficiency is not just shown on simple model proteins but also on proteins such as intrinsically disordered Histatin-5 and metamorphic RFA-H proteins. The different systems considered in this paper are shown in Fig. 1. The computational details of all the systems are presented in Supplementary Table 1.

**Qualitatively better sampling in model proteins with REHT.** We first test the sampling efficiency of our parallel tempering method on well-known model systems such as alanine-dipeptide, TRP-cage, and β-hairpin. The simulations of TRP-cage and β-hairpin were initiated from completely extended conformations of the physiologically relevant zwitterionic state[20] (with charged termini). For comparison against the state-of-the-art method in MD-based sampling, the simulations are also performed with REST2[10,11,14]. The replica mixing of the two simulations in both the systems is shown in Supplementary Figs. 1 and 2.

During the course of the simulations, both the REHT and REST2 methods drive the transition of the unfolded starting conformation to a native folded conformation with almost a perfect match with the experimentally found native structure (0.4 Å root mean squared deviation (RMSD)) as shown in Fig. 2a and Supplementary Fig. 3a. Importantly, we observe faster transitions with REHT between the folded and unfolded basins. The time evolution of RMSD indicates that the REHT method samples the folded structures of the two proteins in less than 100 ns timescales (Fig. 2b) and produces the folded state in 6 out of 12 replicas

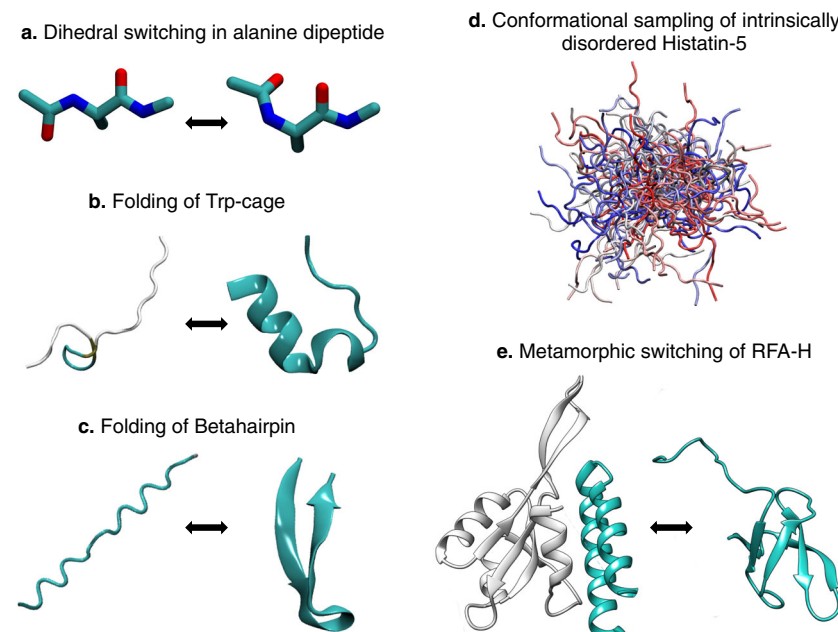

**Fig. 1 List of systems studied. a** Dihedral switching of alanine dipeptide, **b** folding of Trp-Cage from completely unfolded structure, **c** folding of β-hairpin, **d** intrinsically disordered Histatin-5, and **e** metamorphic switching in bacterial RFA-H are explored using the state-of-the art REST2 and the REHT approach designed in this work.

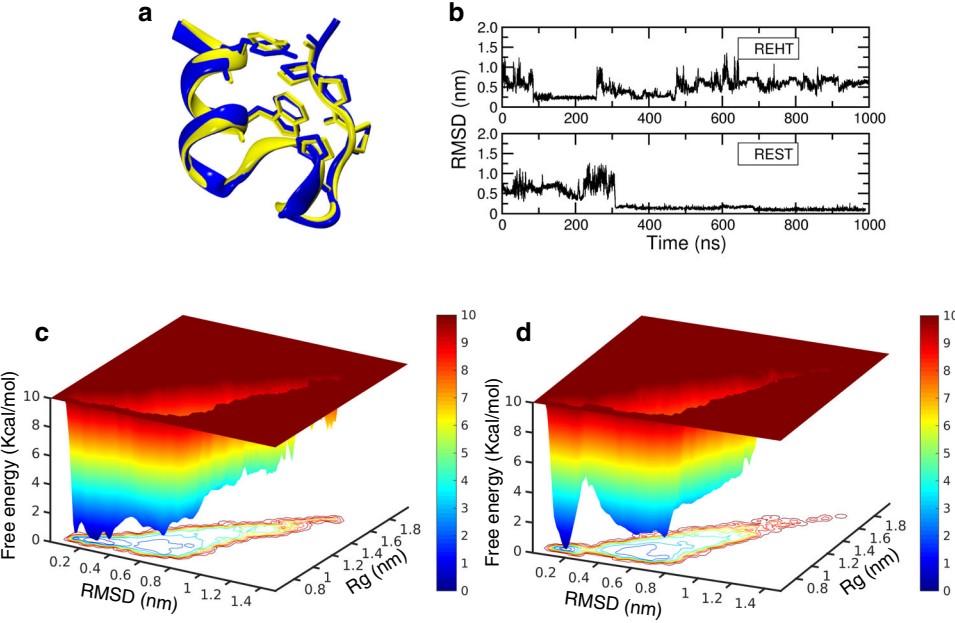

**Fig. 2 Efficiency of REHT in comparison to REST2 in capturing the folding of fast-folding protein, Trp-Cage. a** Structural overlay between the natively folded NMR structure (blue) and the REHT-generated folded structure of Trp-cage (yellow, obtained at the base replica). **b** Time evolution of protein backbone RMSD from the NMR structure along one of the successfully folded replicas of REHT (top) and REST2 (bottom) simulations. The RMSD evolution of other folded replicas are shown in Supplementary Figs. 4 and 5. **c, d** Free energy landscape of Trp-Cage, shown as the function of Radius of gyration (Rg) and RMSD against the NMR structure. The landscape is shown for the ensembles collected at the base replica of **c**) REHT simulation and **d** REST2 simulation.

(Supplementary Figs. 3b, 4). The REST2 simulations sample the folded structure of TRP-cage (Fig. 2b) and β-hairpin (ref. [16]) at around 300 ns. Moreover, only 1–2 replicas out of 8 replicas are independently folded in these simulations (Supplementary Fig. 5 and ref. [16]). Free energy landscapes as a functions of RMSD and radius of gyration for the base replica of TRP-cage simulated with REHT and REST2 are presented in Fig. 2c and Fig. 2d, respectively. The predicted free energy barrier by the REHT

method (~2 kcal/mol) matches closely with the suggested free energy barrier of ~2.1 kcal/mol[21,22], while a larger barrier of about ~6 kcal/mol is observed in REST2 simulations. However, it should be noted that the estimated energy barrier depends on the choice of reaction coordinates used for projecting the landscape. For the ideal reaction coordinates that capture the slowest reaction pathways of protein folding one may need to optimize the reaction coordinates with methods such as path based

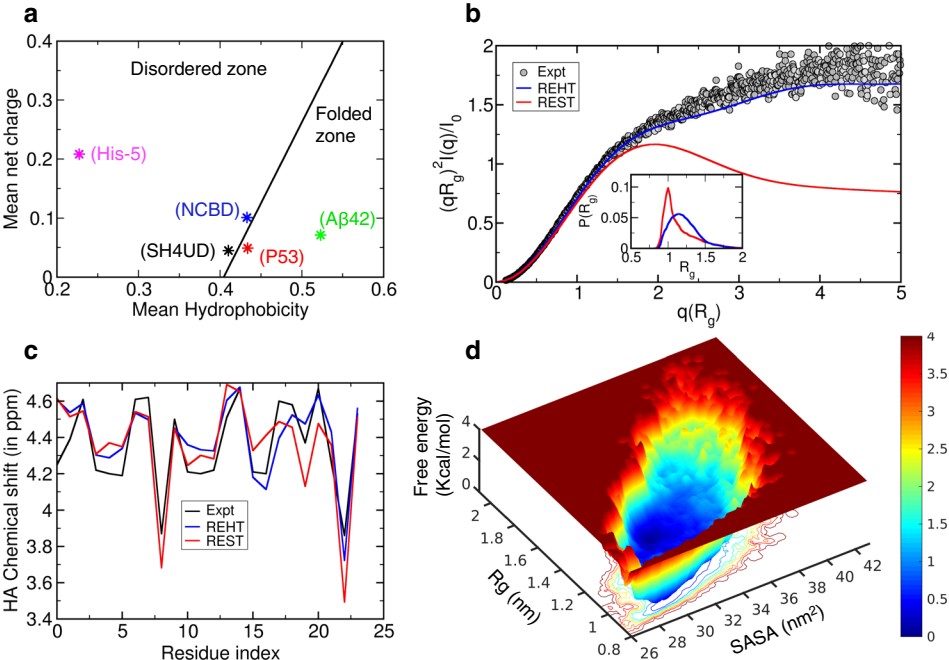

**Fig. 3 Ensemble description of Intrinsically disordered Histatin-5. a** Charge-hydropathy plot showing the uniqueness of Histatin-5 that is located at the disordered zone with lower mean hydrophobicity unlike other successfully studied IDPs which exist at or near the folded zone. **b** Comparison between the experimental (black) and theoretical ensemble-averaged SAXS profiles, represented as a Kratky plot. The theoretical prediction was made for the last 250 ns unweighted trajectories corresponding to the base replica of REHT and REST2 simulations. The distribution of Rg for the ensembles obtained from REST2 (red) and REHT (blue) simulations are shown in the inset. **c** Comparison of ensemble averaged chemical shifts of Hα atoms predicted from the REHT and REST2 simulations (for the same 250 ns trajectory) with reference to the experimental NMR chemical shifts. **d** Weakly-funneled diffusive energy landscape of Histatin-5 explored from REHT simulation is shown as a function of Rg and solvent accessible surface area (SASA).

sampling, and other linear and non-linear combination of methods[23–26]. Also, the generated free energy maps would be meaningful only if the simulations are ergodic. We assessed the ergodicity by comparing conformational distributions of the base replica in two equal halves of the trajectory, similar to that suggested by Dave Thirumalai's group[27] and Bruce Berne's group[10]. The results suggested that the REHT converges faster than the REST2. (See Supplementary Figs. 6, 7 and Supplementary Note 1)

The expedited sampling of the "slow" reaction coordinate (collective variable) is also visible in alanine dipeptide, the quintessential model system for barrier crossing events in biophysical advanced sampling methods development field. We discuss this in detail in Supplementary Note 2 under the heading "Dihedral switch in Alanine dipeptide". In short, we show that the relatively slow transitioning ϕ-angle of the Ramachandran map is frequently sampled in the REHT method (Supplementary Figs. 8, 9) with a larger number of replicas (4 out of 5) exhibiting this transition as well.

**Ensemble sampling of flexible IDPs with high charge and low hydrophobicity.** Accurate conformational sampling of IDPs is a major challenge in the field of molecular simulations where conventional forcefields do not seem to accurately reproduce the properties of IDPs. Several solutions toward fixing the forcefield transferability issues have been suggested and applied extensively to IDPs[28–31]. What stands out in these improvements is the necessity to have a balanced protein–water interactions besides the required changes in the parameters of proteins[32,33]. Also, rare-event sampling methods are frequently used in conjunction with these improved force-fields to faithfully capture the conformational landscape of IDPs as shown for P53[34], α-synuclein[35], islet amyloid polypeptide[36], amyloid-β[37,38], and NCBD IDPs[39].

Recently, REST2 in combination with the IDP-specific forcefield (Amberff03ws) and water model (TIP4P/2005s) was used to successfully yield experimentally consistent ensemble in SH4UD[14].

Despite these successes, there are set of IDPs on which the simulations data show striking deviations from the experimental results. For instance, in a 24-residue long antimicrobial peptide Histatin-5 (His-5), the simulation-generated ensemble deviates substantially from the experimental data from circular dichroism (CD), nuclear magnetic resonance (NMR), and small angle X-ray scattering (SAXS) measurements[40,41]. His-5 is not accurately sampled even with the state-of-the-art forcefields[40,41] suggesting their lack of transferability across IDPs with diverse range of charge and hydropathy characteristics. We illustrate the charge-hydropathy (CH) plot of various IDPs in Fig. 3a. The plot clearly shows that the successfully characterized proteins (p53, AB42, NCBD, SH4UD) fall at or near the line separating the folded and unfolded region indicating the pre-molten globular nature of these IDPs. His-5 on the other end is located at the far end of CH plot with very minimal hydrophobicity and higher charges.

We sampled the conformational landscape of His-5 using REHT and REST2 (see Supplementary Fig. 10 for mixing of replicas) and verified the accuracy of the generated ensemble against various experimental ensemble average properties. To start with, we evaluated the secondary structure content of this peptide by analyzing the backbone dihedrals of all the residues. The CD data[42] reveals that His-5 prefers to form polyproline II (PPII) structures and the PPII propensity is lost at higher temperatures. The Ramachandran map for His-5 (Supplementary Fig. 11) indicates the most probable occurrence of PPII structures, which is generally not recapitulated in the conventional IDP sampling methods[42]. Interestingly, enhancing the sampling either by REHT or REST2 recapitulates the propensity of PPII

structures of His-5 in qualitative agreement with the CD measurements. Furthermore, the temperature dependent loss of PPII structure is also captured correctly (Supplementary Fig. 11).

We also calculate the nanoscale structural properties of His-5 measured experimentally by means of SAXS that describes the overall size and shape of a protein in solution. The experimental SAXS data[43] for His-5 at room temperature and at neutral pH was obtained from Prof. Marie Skepo's laboratory in Sweden. The results are depicted in Fig. 3b as dimensionless Kratky representation, which qualitatively assesses the compactness and flexibility of a protein. This can be obtained from the form factor using the following equation: $(qRg)^2 I(q)/I(0)$ and plotted against $qRg$. In case of compact well-folded proteins, the Kratky plot exhibits a bell-shaped peak at low-q regime and converges to the q-axis at high-q regime. Conversely the disordered proteins, depending on the degree of compactness, flexibility of the chain and the presence of structured regions show different curves. For the completely expanded or fully unfolded proteins the intensity at high-q region exhibits a plateau, that may be followed by further rise in some cases[44,45]. As shown in the figure, the Kratky plot of His-5 obtained from the experiment exhibits the signature of highly flexible and extended IDP. The SAXS profile from REHT ensemble data (Fig. 3b) matches closely with that of the experiment, thereby reinforcing its fidelity in constructing the IDP ensemble accurately. We believe that the conformations of IDPs that lie significantly away from the CH border area are far more affected by the hydration-induced stability. In such cases, REST2 is not suitable since it does not treat the surrounding water and hence produces a more compact protein configurational ensemble as evident from both the SAXS profile and Rg plot (Fig. 3b: inset). We also compare the chemical shifts of hydrogen atoms linked to the Cα predicted from the REHT and REST2 methods with that of the NMR chemical shifts data[46] (Fig. 3c). The chemical shifts predicted from REHT matches better with the experiments than the REST2 method.

After verifying the REHT-generated ensemble with the experimental data from CD, SAXS, and NMR measurements, we reconstitute the free energy map of His-5 at the room temperature as a function of various structural parameters (Fig. 3d, Supplementary Fig. 12). The free energy map indicates rather a flat-bottom low-energy landscape across a wide range of Rg (from 1.0–1.8 nm) and SASA values (28–34 nm$^2$). The results indicate that under solution conditions, the peptide prefers to exist in a completely disordered conformation. Such a large disorderliness endows them with an ability to bind diverse targets as confirmed from a proteomic analysis[47]. We also traced the residual α-helical propensity as observed in the non-aqueous solutions and model lipid vesicles that couples to its candidacidal functioning[46,48,49]. However, we did not observe any trace of helicity in the solution ensemble (Supplementary Fig. 13), suggesting that the helical transition may be adapted in an induced fit manner upon association with the membrane.

In contrast the REST2-generated ensemble produces a funneled free energy landscape with restriction at a compact Rg value (Supplementary Fig. 12). Moreover, to attain a converged sampling distribution, the REST2 would require 12 fold more CPU time than REHT (Supplementary Figs. 14–16 and Supplementary Note 3). While comparing the free energy maps between REHT and REST2, one can easily speculate the amount of heterogeneity added to sampling by the REHT method. Further, we quantitatively estimated the heterogeneity of REST2 and REHT ensemble at the base replica by measuring pairwise distances between conformations (Supplementary Note 4)[50,51]. The measurements (Supplementary Fig. 17), however, account more heterogeneity for the REST2, in spite of the confined sampling of compact states. These conflicting results needed to be reconciled, and we anticipated that one of the possible explanations for this inconsistency could have its origin in the "heterogeneous compact structures" that the REST2 simulations sample for the His-5 ensemble.

To inspect this, we generated a 2-dimensional map of ensemble based on the pairwise conformational distances using multi-dimensional scaling (MDS). MDS is particularly useful for visualizing the distance matrix in low dimensional space while preserving the distances between objects as much as possible. The conformational map along the MDS coordinates is presented in Fig. 4 for both REST2 and REHT. Coloring each conformation with the respective Rg value (Fig. 4a, b) shows that there are indeed more and diverse compact states in REST2 (than REHT) that spread all over the space. At the same time the replica index-based coloring reveals that the distinct compact conformations are from different replicas that occupy non-overlapping and maximally separated region (for replica number 2, 3, 4 and 7), as shown in Fig. 4c. This result clearly shows that in REST2 the enhanced heterogeneity arises in the base replica by the virtue of exchange of conformations between replicas that are stuck at different local basins of compact structures. (The conformational trap of individual replica in REST2 can also be seen from Supplementary Figs. 15 and 18). On the other hand, in REHT, the contributions from independent replicas are not significantly different or distant from each other and within a single replica they sample a diverse space (Fig. 4d).

The exhaustive sampling of heterogeneous conformations enables the mixing process in REHT that otherwise results in eddies formed at the higher and lower zones of the replicas as seen in REST2. Essentially the exchanges between replicas take place in zones—and interzonal exchanges do not take place, which affects the overall sampling drastically. This behavior is readily apparent in large complex proteins like that of metamorphic RFA-H as will be shown in the next section.

**Deciphering the transition intermediates of metamorphic protein (RFA-H).** Unlike the weakly-multi-funneled landscapes generally found in IDPs, metamorphic proteins have deep multi-funnels with each low energy basin representing a distinct well-folded conformation. These conformations switch upon being signaled to perform different functions. Among the many members of metamorphic proteins known till date, RFA-H possesses highly divergent folds with completely different secondary structures. RFA-H is a bacterial regulatory protein, in which its C-terminal domain alternates between all α-helical and all β-sheet conformations based on presence or absence of inter-domain contacts respectively (Fig. 5a)[52]. While in helical conformation, it activates the transcriptional elements, in β-sheet conformation it recruits a ribosomal protein and thereby couples to translation[53,54]. Transition between these alternate conformations generally involve unfolding from one conformation and refolding to the other conformation[55]. Exploring these protein landscapes by MD simulations is highly challenging and has been attempted rarely by utilizing biasing techniques such as targeted MD or assisted GO model[56–58].

We set to explore the conformational metamorphosis of RFA-H in explicit water using our unbiased REHT simulation. The same was attempted using REST2 simulation as well, which did not yield proper mixing of replicas even at 250 ns of simulation time per replica (Supplementary Fig. 19) and hence was dropped. REHT started showing mixing between replicas by around 100 ns and the runs were extended to 1000 ns/replica for the 25 replicas. The experimentally obtained α-helical state (PDB ID: 2OUG) of the C-terminal domain (CTD (residue number 115-162)) was used as starting structure (Fig. 5b). Unlike the previous studies

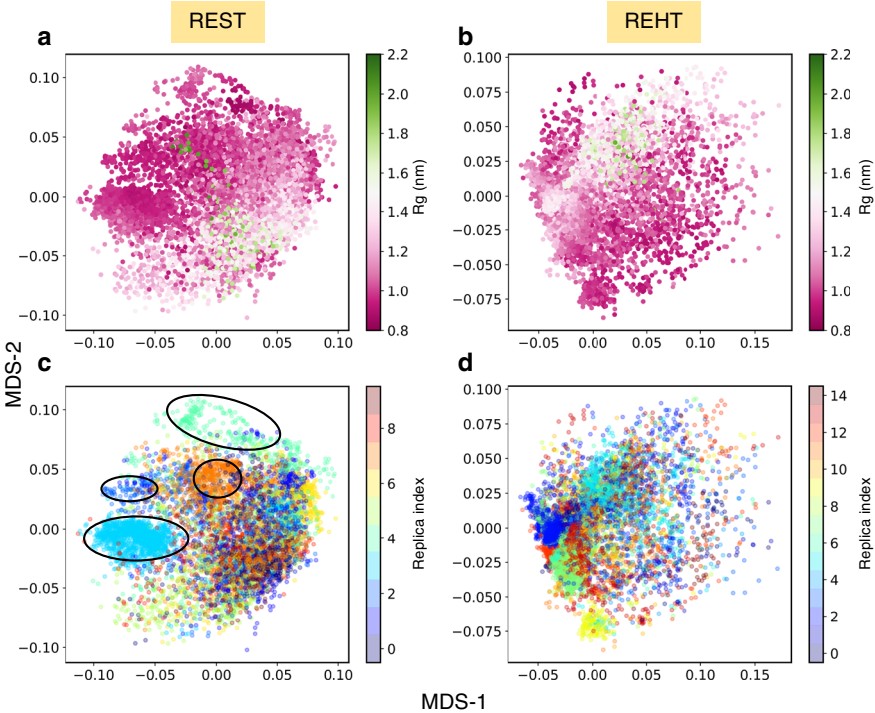

**Fig. 4 Configurational map of His-5 ensemble based on pairwise-dissimilarity of conformations shown across 2-dimensional MDS axes.** In **a**, **b** the conformations of REST2 and REHT ensembles are colored by the respective Rg values and in **c**, **d** they are colored by replica index. Maximally separated non-overlapping clusters in REST2 as marked in **c** indicates the trapping of conformations in independent replicas that are eventually exchanged to base replica.

using targeted MD[56,58] or assisted GO model with replica exchange tunneling[57], the REHT method does not embed in itself any information of the target state. In spite of that, the simulations spontaneously transform the RFA-H$_{CTD}$ from helical structure to the β-barrel conformation. A secondary structure analysis of the ensemble generated at the lowest temperature (310 K) replica reveals that this transformation occurs within 300 ns (Supplementary Fig. 20). The residues that were shown to form β-sheets agree well with the NMR-generated structure (PDB ID: 2LCL). The structural overlay of a snapshot obtained from the simulation on to the NMR structure is shown in Fig. 5b and their RMSD is calculated to be 3.4 Å. It should be appropriate to note here that the previous unbiased MD simulation, with implicit solvation model using the effective energy function EEF1[59], fails to produce the β-barrel spontaneously. Additional refinement with an explicit solvent simulation was required to arrive at the barrel structure with ~5 Å RMSD with respect to the experimental structure[60].

To further validate our results, we predicted the backbone chemical shifts of the simulation-generated ensemble obtained from the room temperature replica. The last 500 ns timeframes without any reweighting or constricted selection was used for this analysis. The predicted Cα chemical shifts were compared with those measured from NMR spectroscopy (Fig. 5c). The figure clearly demonstrates a striking agreement between the predicted and measured chemical shift values. Following the experimental validation, we traced for the molecular details of the fold switching mechanism from our atomic-resolution trajectory, the information that cannot be obtained directly from the experiments. Toward this, we plotted the free energy map as a function of RMSDs with respect to the experimentally found helical and beta-barrel structures (Fig. 5d). The map reveals dual-basins, each corresponding to the different folds of the RFA-H, where the basins are separated by a free energy barrier of about 5 Kcal/mol. This barrier corresponds to the α → β transition while not

favoring the reverse transition. This argument is based on our observation with increasing simulation time, the population of β conformation shows an increase suggesting a (non-converged) deeper basin for the β structure. (Supplementary Fig. 21) For the reverse transition β → α, experiments indicate that the N-terminal interaction is important[52]. To reinforce this statement, we simulated additional simulation initiated from the NMR-obtained β-structure, which show stable β-barrel structures (Supplementary Fig. 22), and is consistent with the NMR observations[52].

While the free energy basin corresponding to the α-helical structure exhibits a funnel like architecture, that of the β-barrel structure looks like a rugged landscape indicating possible heterogeneity in that state. The heterogeneity likely builds up the entropic barrier and adds to the complexity of the free energy map and thereby limits the spontaneous transition from one state to the other upon using conventional advanced sampling methods. REHT shows promising results in crossing both the enthalpic and entropic barriers and therefore allows for studying the complex conformational transitions of IDPs and "transformer" proteins such as RFA-H. The method also opens up avenues to study other transformer proteins such as chemokine lymphotactin protein, Mad2 spindle checkpoint protein, CLIC1[4].

**Modulated tempering of water self-interaction facilitates entropic barrier crossing and promotes conformational search.** Our method has its origin from REST2 where we try to solve the issues arising out of "cold solvent" while keeping the computational requirements tractable. REHT shows remarkable improvements in sampling converged and experimentally consistent ensembles with using the same forcefield. Also, both methods impose no bias in the base replica either directly or indirectly from the high temperature replicas that produces identical equilibrium ensemble conditions (Supplementary Fig. 23). In spite of that, the striking difference in the quality of

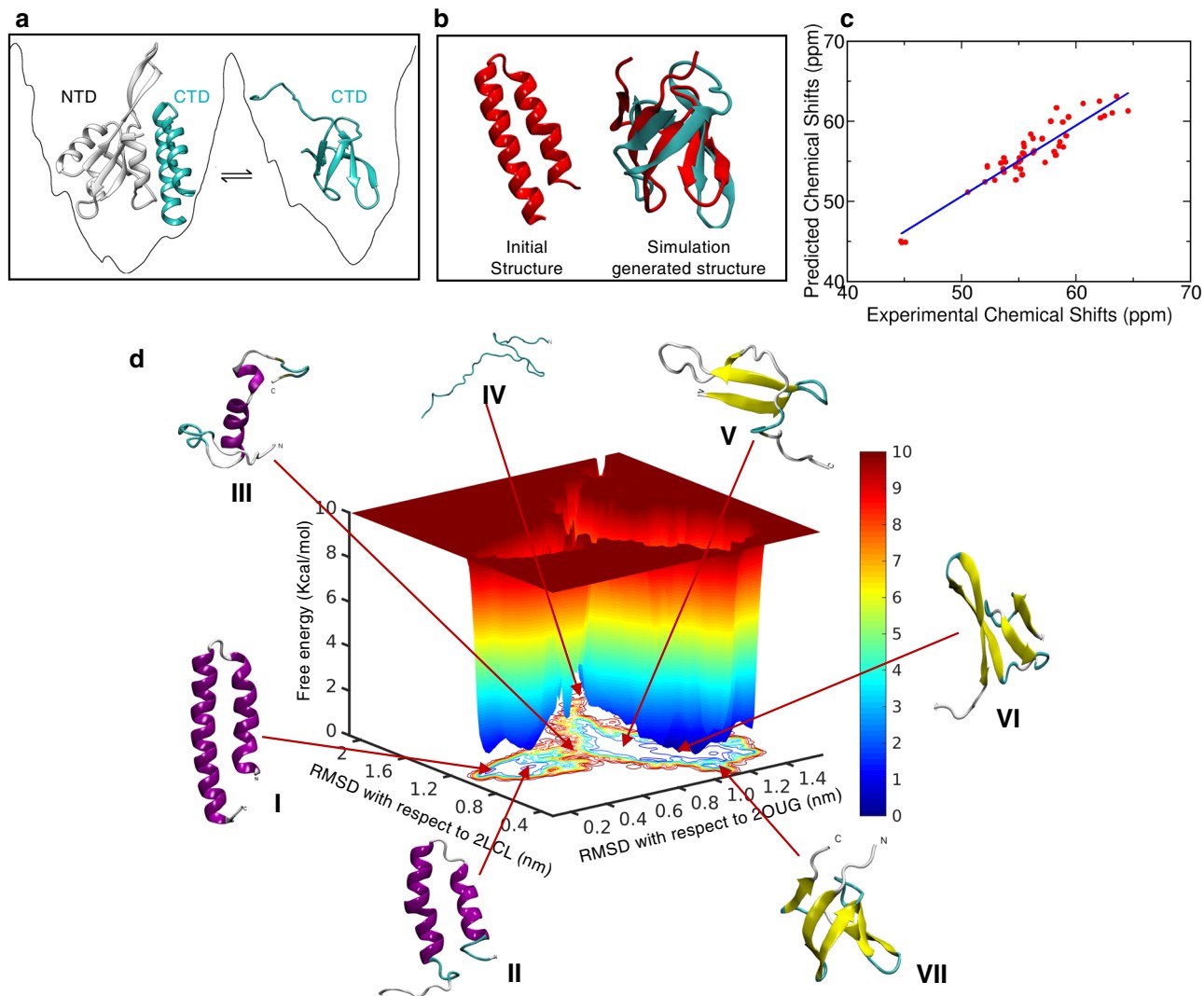

**Fig. 5 Conformational metamorphosis of RFA-H. a** Schematic representation of the two native structures resolved in experiments, the α-helical fold of CTD in presence of NTD and β fold in isolated CTD. **b** The initial and final structures of the REHT simulation. The helical state of CTD while deleting the NTD was chosen as the starting structure (left figure in **b**). Structural superposition of the final simulation generated β barrel structure (red) over the experimental β structure (cyan) is shown on the right side of (**b**). **c** Validation of simulation generated ensemble by comparing the predicted Cα chemical shifts with the experimental shifts. The fitted linear regression (blue line) indicates a precise match between the two sets of data. **d** Dual basin free energy landscape of RFA-H shown as a function of RMSDs from the experimentally found α-helix (2OUG) and β-sheet (2LCL). The conformations at each of the basins, all α-helical and all β-sheet (I and VII), early structure with the loss of helicity at both the termini (II), intermediate partially unfolded structure with residual α-helical content (III), and metastable structures with open β-sheets (V and VI) are shown along the landscape. Note that the completely unfolded structure (IV) is at the other side of the dual basin and the transition of α → β structure does not need to step through the completely unfolded structure unlike that of metamorphic lymphotactin[55].

sampling is due to the way water self-interaction is treated in REHT and REST2. The comparison of hydration shell properties for the two methods shed some very interesting and revealing insights into the possible origins of differences in the sampling and substantially shortened round trip times for REHT as compared to REST2.

It should be noted that the trajectory at the base replica is not continuous with respect to time (the system coordinates are exchanged across different replicas at every successful exchange). Therefore, comparing the hydration shell dynamics of base replicas would give rise to incomprehensible results. Hence, we chose to do the same analysis on the time-continuous trajectories for all replicas. We extracted and de-mixed the trajectories (1 ns each) from various time points of the simulations (For example: $t = 50$ ns, 100 ns, 150 ns and up to 1000 ns). We performed the calculation of water reorientation dynamics in the hydration-shell

for all the trajectories using the MDAnalysis tool[61]. In the Supplementary Fig. 24, we show a representative set of curves on the water reorientation decay in one of the replicas (replica 0 at various time points) of Trp-cage simulated with REHT method.

The curves can be represented by a biexponential decay ($C_t = A \, exp^{-x/\tau_1} + B \, exp^{-x/\tau_2}$) with a set of fast ($\tau_1$) and slow ($\tau_2$) decay constants. We extracted these relaxation decay constants for all the sampled time-continuous trajectories of TRP-cage and His-5 with REHT and REST2 simulations and plotted their cumulative distribution in Supplementary Fig. 25 and Fig. 6. The distribution of fast component $\tau_1$ (~1 ps) that is considered relatively independent of temperature[62] exhibit similar trends in both REST2 and REHT simulations for both Trp-cage and His-5 (Supplementary Fig. 25). On the contrary, the slow component $\tau_2$ is thermally activated and its distribution is noticeably different for REST2 and REHT methods (Fig. 6a, b). The activation energy

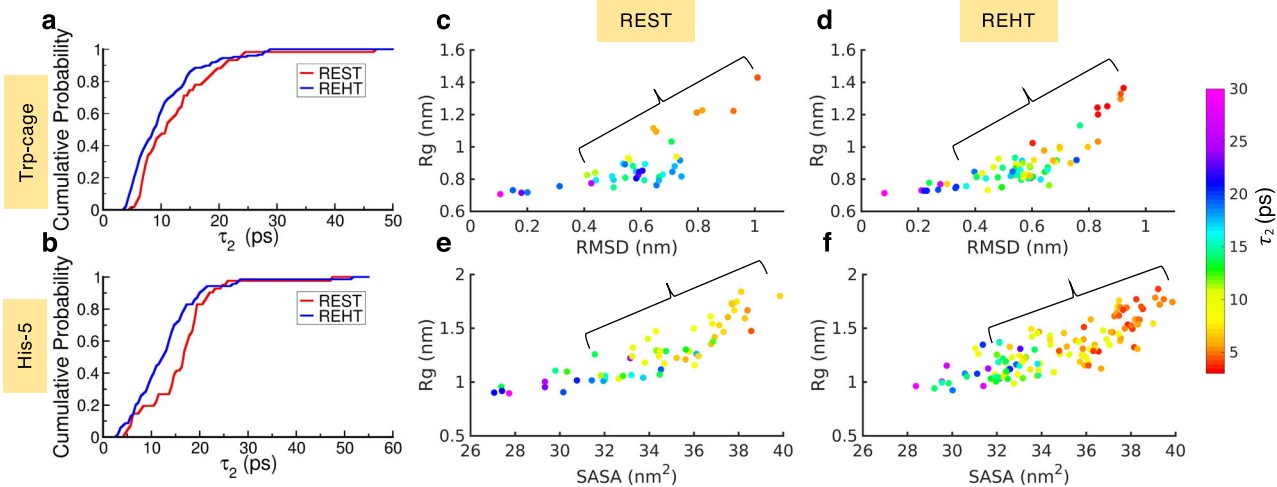

**Fig. 6 Extent of hydration dynamics and its tight coupling with the conformational space sampling.** In figure **a**, **b** we showed the cumulative distribution of the slow component of water orientational relaxation decay ($\tau_2$) in REST2 (red) and REHT (blue) simulations for Trp-cage (**a**) and His-5 (**b**). In **c**–**f** we projected the conformational space of Trp cage (across RMSD to the native NMR structure and Rg) and His-5 (across SASA and Rg) with the conformations color coded by the $\tau_2$ value as indicated in the color bar. The region with significant entropic barrier is highlighted in all the systems. Since there are a lesser number of replicas in REST2 in comparison to REHT (8 vs 12 in Trp-cage, 10 vs 15 in histatin-5), the number of points analyzed also became lesser in REST2.

for the slow relaxation ($\tau_2$) is reduced in REHT due to differential treatment of solvent. This yields a comparatively faster relaxation than REST2 and in some sense overcomes the "cold solvent" bottlenecks that is characteristic of REST2.

The faster decay in the water orientation facilitates the rewiring of protein–water H-bonds allowing for the faster conformational transitions (Supplementary Fig. 26)[63]. Thus, the water relaxation is tightly coupled with the protein conformations through the protein–water H-bonds. The orientation relaxation decay of water is slower in ordered conformations and faster in extended disordered conformations of protein. We show this coupling by plotting the protein conformations colored with the $\tau_2$ value of water orientation relaxation in Fig. 6c–f. The water relaxation, in both the REST2 and REHT simulations, is restricted in compact and ordered states to the same extent. However, interestingly the major difference originates from the landscape where the entropic barrier dominates, which corresponds to the unfolded/extended disordered states to the intermediate state (indicated by braces). This region shows relatively faster relaxation in REHT than REST2 thereby providing a smoother surface for the conformational transitions from unfolded to intermediate state. We believe that this would result in a reduced entropic barrier between the unfolded and intermediate states in REHT than REST2. Further down the hill toward the compact state, the water relaxes much slower and enthalpic interactions, which tends to become stronger with the formation of inter residue contacts (in case of folded proteins), generally drives the process of folding. The IDPs in which the enthalpic compensation is weaker (due to low hydrophobicity and high net-charge), favors disordered states.

Taken together, the results provide a compelling evidence that the REST2 is stuck at entropic barriers. By optimally heating up the water, our method overcomes this barrier efficiently and allows for the rapid generation of more faithful ensemble.

## Discussion

In this work, we design a scaled Hamiltonian that differentially treats the protein and solvent interactions as a function of temperature in the replica-exchange framework. Essentially, the design of the Hamiltonian in our method (REHT) allows faster

decay of water orientation dynamics, which in turn enhances the conformational sampling for proteins. We find that the accelerated thermodynamic sampling in REHT compensates for the additional computational cost incurred by the moderately higher number of replicas due to inclusion of water interactions in the redesigned Hamiltonian for replica exchanges. The high-resolution structural ensemble for a variety of proteins produced through our REHT simulations agree excellently with the ensemble average observables obtained from NMR, SAXS, CD, and other biophysical experiments without the need of any reweighting. The method is particularly suited for highly flexible IDPs such as His-5 where solvation dynamics stabilize and drive the coexistence of multiple degenerate extended states of the peptide on a free-energy surface that have multiple shallow basins. Large free energy barriers in metamorphic proteins, with multiple well-defined basins having diversely folded conformations, are usually not easily surmountable through conventional methods. We show that REHT is capable of sampling across the barriers of metamorphic proteins, deduce possible transition intermediates and solve their conformational ensemble. We believe that when appropriately coupled with the known experimental data, the high-fidelity structural ensemble information from the unconstrained REHT-simulations can be effectively used to sample and if needed further minimize the residual with experimental observables using integrative modeling framework[64–66].

## Methods

**Derivation of REHT method.** We provide the theoretical background about the different versions of replica exchange methods in Supplementary Note 5. Here, we derive the detailed balance exchange condition for the REHT method, with a specially designed Hamiltonian that differentially treats solvent and solute across a temperature range.

Let us consider the collection of replicas simulated in replica exchange simulation as {$X_1, X_2 \ldots X_n$}. In general, the replicas differ in temperature ($Ti$) while use an identical Hamiltonian function (Temperature replica exchange) or vice versa (Hamiltonian replica exchange). In our hybrid approach, we change both the temperature as well as the Hamiltonian across the replicas. Hence the replicas can be denoted as {$X_m, H_m(X_m), T_m$}, where $X_m$, $H_m(X_m)$, and $T_m$ respectively are configuration, potential energy function and temperature of replica $m$.

Since the replicas are non-interacting, the equilibrium probability of this larger ensemble can simply be obtained by the product of Boltzmann factors of each

replica.

$$P_{REM}(X) = \prod_{i=1}^{N} \frac{1}{Z_i} \exp(-\beta_i H_i(X)), \qquad (2)$$

where $\beta_i$ denotes the inverse temperature ($1/k_B T$) and $Z_i$ represents the configurational partition function.

Consider an exchange of configurations between a pair of replicas m and n.

$$\{X_m, H_m(X_m), T_m\} \rightarrow \{X_n, H_m(X_n), T_m\}$$
$$\{X_n, H_n(X_n), T_n\} \rightarrow \{X_m, H_n(X_m), T_n\}$$

The probability density of states before and after the exchange are given as,

$$\rho_{before} = [\exp -[\beta_m H_m(X_m) + \beta_n H_n(X_n)]]/Z$$
$$\rho_{after} = [\exp -[\beta_m H_m(X_n) + \beta_n H_n(X_m)]]/Z$$

The detailed balance condition and the corresponding transition probability for this exchange is given by,

$$\rho_{before}\Pi(X_m \rightarrow X_n) = \rho_{after}\Pi(X_n \rightarrow X_m)$$
$$\Pi = \frac{\rho_{after}}{\rho_{before}} = \frac{\Pi(X_m \rightarrow X_n)}{\Pi(X_n \rightarrow X_m)} \qquad (3)$$

Imposing the detailed balance condition, where the reverse exchange is allowed with equal probability,

$$\Pi = \frac{\exp -[\beta_m H_m(X_n) + \beta_n H_n(X_m)]}{\exp -[\beta_m H_m(X_m) + \beta_n H_n(X_n)]} \qquad (4)$$

$$= \exp -[\beta_m H_m(X_n) + \beta_n H_n(X_m) - \beta_m H_m(X_m) - \beta_n H_n(X_n)] \qquad (5)$$

$$= \exp(-\Delta_{nm}) \qquad (6)$$

where $\Delta_{nm}$

$$= \beta_n[H_n(X_m) - H_m(X_m)] + \beta_m[H_m(X_n) - H_n(X_n)] \\ - (\beta_n - \beta_m)[H_n(X_n) - H_m(X_m)] \qquad (7)$$

With the Metropolis criteria, the probability of accepting the exchange $X_m \rightarrow X_n$ becomes,

$$\Pi(X_m \rightarrow X_n) = \begin{cases} 1 & if\ \Delta_{nm} \leq 0 \\ \exp(-\Delta_{nm}) & if\ \Delta_{nm} > 0 \end{cases} \qquad (8)$$

The raw derivation from first principle is given in the following github link (https://github.com/codesrivastavalab/ReplicaExchangeWithHybridTempering/blob/master/REHT-ShareFiles/reht-derivation.pdf).

We started from a deformed Hamiltonian identical to the one used in the REST2 approach.

$$H_m = \lambda_m H_{pp} + \sqrt{\lambda_m} H_{pw} + H_{ww} \qquad (9)$$

where $\lambda_m$ denote the scaling factor of the Hamiltonian in replica m. $H_{pp}$, $H_{pw}$ and $H_{ww}$ are the potential function for protein-protein, protein–water and water–water interaction energies.

Recombining the energy difference due to temperature bias (Eq. (7)) and the Hamiltonian bias as in Eq. (9), $\Delta_{nm}$ for REHT becomes,

$$= \beta_n \left[ (\lambda_n - \lambda_m)\left(H_{pp}(X_m) + \left(\sqrt{\lambda_n} - \sqrt{\lambda_m}\right)H_{pw}(X_m)\right) \right]$$
$$+ \beta_m \left[ (\lambda_m - \lambda_n)\left(H_{pp}(X_n) + \left(\sqrt{\lambda_m} - \sqrt{\lambda_n}\right)H_{pw}(X_n)\right) \right] \qquad (10)$$
$$- (\beta_n - \beta_m)\left[ \begin{array}{c} \lambda_n H_{pp}(X_n) + \sqrt{\lambda_n} H_{pw}(X_n) + H_{ww}(X_n) - \lambda_m H_{pp}(X_m) \\ - \sqrt{\lambda_m} H_{pw}(X_m) - H_{ww}(X_m) \end{array} \right]$$

Coefficient of $H_{pp}$:

$$= \beta_n[(\lambda_n - \lambda_m) + (\beta_n - \beta_m)\lambda_m]H_{pp}(X_m)$$
$$+ \beta_m[(\lambda_m - \lambda_n) - (\beta_n - \beta_m)\lambda_n]H_{pp}(X_n) \qquad (11)$$
$$= -(\beta_n \lambda_n - \beta_m \lambda_m)\left[H_{pp}(X_n) - H_{pp}(X_m)\right]$$

Coefficient of $H_{pw}$:

$$= \left[\beta_n\left(\sqrt{\lambda_n} - \sqrt{\lambda_m}\right) + (\beta_n - \beta_m)\sqrt{\lambda_m}\right]H_{pw}(X_m)$$
$$+ \left[\beta_m\left(\sqrt{\lambda_m} - \sqrt{\lambda_n}\right) - (\beta_n - \beta_m)\sqrt{\lambda_n}\right]H_{pw}(X_n) \qquad (12)$$
$$= -\left(\beta_n\sqrt{\lambda_n} - \beta_m\sqrt{\lambda_m}\right)\left[H_{pw}(X_n) - H_{pw}(X_m)\right]$$

Coefficient of $H_{ww}$:

$$-(\beta_n - \beta_m)[H_{ww}(X_n) - H_{ww}(X_m)] \qquad (13)$$

Combining Eqs. (11), (12) and (13),

$$\Delta_{nm}(REHT) = -\left[ \begin{array}{c} (\beta_n \lambda_n - \beta_m \lambda_m)\left[H_{pp}(X_n) - H_{pp}(X_m)\right] \\ + (\beta_n\sqrt{\lambda_n} - \beta_m\sqrt{\lambda_m})\left[H_{pw}(X_n) - H_{pw}(X_m)\right] \\ + (\beta_n - \beta_m)[H_{ww}(X_n) - H_{ww}(X_m)] \end{array} \right] \qquad (14)$$

**List of systems studied and preparation of atomistic model**. Figure 1 and Supplementary Table 1 contain information about the diverse sets of proteins used in this work.

The initial unfolded structure of Trp-Cage and β-hairpin (C terminal hairpin of B1 domain in protein G) were obtained by simulating the corresponding folded NMR structures (PDB ID: 1l2Y and 1lE3, respectively) at 600 K temperature for 10 ns. For His-5, the unfolded structure was built using VMD protein builder by feeding in the sequence information (DSHAKRHHGYKRKFHEKHHSHRGY). Simulation of RFA-H C-terminal domain was initiated from the α-helical conformation (PDB ID:2OUG). All the proteins (system 1–5 in Fig. 1) were solvated in a cubic box with a minimum distance of 1.2 nm from the surface of the protein (in case of RFA-H 1.5 nm is used). A 3-site rigid TIP3P water model was used for all the systems. The systems were also neutralized with physiological concentration of NaCl (0.15 M). For the topological parameters of fast-folding proteins, we used Amberff14SB in order to faithfully compare the efficiency with earlier REST2 and gREST simulations[16], whereas for all the rest of the systems (alanine dipeptide, His-5 and RFA-H), Charmm36m[29] force field was used.

**Details of MD simulation**. The modeled proteins solutions were initially energy minimized using steepest descent algorithm for 50,000 steps to avoid any poor contacts. The minimized structure was then thermalized and equilibrated sequentially in NVT and NPT ensembles each for 2 ns. The protein and the solvent were coupled separately to the target temperatures using a modified Berendsen thermostat. The pressure was set at 1 bar using Parrinello-Rahman barostat. The final production simulation was performed in NVT ensemble using Nose-Hoover thermostat. A cut-off of 1 nm was used for calculating the electrostatics and VdW interactions. Particle Mesh Ewald was used for long-range electrostatics. Leap-frog integrator with a time step of 2 fs was used to integrate the equations of motions. All the hydrogen atoms were constrained using LINCS algorithm. The simulations were performed using Gromacs-2016.5 patched with Plumed-2.4.1. All the parameter files are available in the GitHub repository (https://doi.org/10.5281/zenodo.4361714)[67].

**Replica exchange parameters**. For all the simulations, we ran both the REHT scheme designed in this work as well as the state-of-the-art REST2 method. The two methods differ in the way the protein and water molecules are treated in the Hamiltonian. REST2 tempers only the protein but keeps the water at room temperature by scaling the potential energy function of solute. The bath temperature used in REST2 is constant across all the replicas. On the other hand in REHT method, the bath temperature is raised mildly up to 340 K as we go up on the replica ladder. This sufficiently heats up the water molecules allowing for its efficient dynamics in REHT method. For enhancing the protein dynamics, the potentials of the protein solute are scaled up to a maximum factor (λ) of ~0.5. To ensure the faithful comparison, the overall effective temperature realized on the protein is identical in both the simulations (Supplementary Table 1). However, due to the extra degrees of freedom due to treatment of the water, the number of replicas used is moderately higher for the REHT method than for the REST simulations. More details and the scripts for all the input preparation (including the scaling of Amber14SB and charmm36m forcefields) are given in the github link: (https://doi.org/10.5281/zenodo.4361714)[67].

**Theoretical SAXS analysis**. Theoretical SAXS profile for the HIS-5 ensemble was predicted using CRYSOL-2.8.4[68] that calculates orientationally averaged scattering pattern using multipole expansion from atomic coordinates while considering the solvation shell using spherical harmonics. The predicted value is compared against the experimental form factor of His-5 obtained at neutral pH from Skepo et al.[43].

**Theoretical analysis of chemical shifts**. The backbone chemical shifts of His-5 and RFA-H atomic coordinates were calculated using SPARTA+ (v2.90)[69], that works based on artificial neural network. To validate our simulation, the predicted shifts were compared against the experimental NMR chemical shifts as obtained from BMRB database (https://bmrb.io) entry 17615[52] and from literature[46].

**Structural analysis**. Other structural parameters such as root mean squared deviation (RMSD), radius of gyration (Rg) and backbone dihedral angles were analyzed using the respective Gromacs utilities. Analysis of protein secondary structures was performed using DSSP tool (v2.2.1). VMD was used for visualizing the trajectory and rendering of protein cartoon images.

**Analysis of hydration dynamics**. The dynamical properties of the hydration shell were calculated by means of water orientational relaxation and hydrogen bond life-time decay as implemented in MDAnalysis package (v 0.19.2)[61]. A Continuous trajectory with respect to simulation time was used for this analysis. The orientational relaxation essentially indicates the rotational freedom of water molecules in a solvation shell. This is measured by computing the second order rotational autocorrelation function of two vectors: one along the OH bond and the other along the dipole moment of water, as given as: $C_{2\hat{u}}(\tau) = P_2\left(\hat{u}_{t_0} \cdot \hat{u}_{t_{0+\tau}}\right)$, where, $P_2(x)$ is the second Legendre polynomial and $\hat{u}$ is the unit vector. Similarly, we measure the persistence of interaction between the protein and the water molecules by computing the auto-correlation function of Hydrogen bond lifetime, as follows: $C_{HB}(\tau) = \frac{\sum_{ij} h_{ij}(t_0) h_{ij}(t_{0+\tau})}{\sum_{ij} h_{ij}(t_0)}$, where, the H-bond between pair $i$ and $j$ at time $t$, ($h_{ij}(t)$ is considered to be 1 if the geometric distance (<3.5 Å) and angle (>120°) cutoff are met. $h_{ij}(t) = 0$ otherwise. We considered two types of protein–water H-bonds: 'continuous H-bond' when a water molecule is continuously involved in H-bonding and 'intermittent H-bond' when the H-bond is retained with intermittent change of water molecules.

**Calculations of free energy change**. The relative Gibbs free energy of an equilibrium ensemble is computed as a function of two reaction coordinates as follows:

$$\Delta G_{(R1,R2)} = -k_B T \ln \frac{P_{(R1,R2)}}{P_{Max}}, \qquad (15)$$

where $k_B$ represents the Boltzmann constant, $T$ is the temperature. $P_{(R1,R2)}$ denotes the probability of states along the two reaction coordinates, which is calculated using a k-nearest neighbor scheme and $P_{Max}$ denotes the maximum probability. The 3-dimensional representation of the free energy surface was plotted using Matlab.

**Measuring conformational heterogeneity and multi-dimensional scaling**. The calculations of ensemble heterogeneity is described in detail in SI. In short, we estimated the heterogeneity (dissimilarity or distance between conformations) by measuring pair-wise cosine distance[50] and q-factor[51] between conformations. The pair-wise conformational distances ($n(n-1)/2$ cosine distance values) are then embedded in 2D configurational map of $n$ conformations using multi-dimensional scaling such that the original pair distances are best preserved[70].

**Reporting summary**. Further information on research design is available in the Nature Research Reporting Summary linked to this article.

## Data availability

Data supporting the findings of this manuscript are available from the corresponding author upon reasonable request. The raw data underlying line graphs and scatter plots of the current study are also included along with this article as txt files in a folder on the github repository (https://doi.org/10.5281/zenodo.4361714)[67].

## Code availability

The details of REHT simulation set-up and scripts for generating the input files are available at the same github repository, https://doi.org/10.5281/zenodo.4361714 [67].

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

## Acknowledgements

The authors are grateful to the SciNet HPC Consortium, ComputeCanada[71,72] for the computational resources and Prof. Marie Skepo for the Histatin-5 SAXS data. A.S. and A.R. thanks Dr. Jagannath Mondal, Dr. Ashok Sekhar, and Dr. Debostuti Ghoshdastidar for their valuable comments and suggestions on the manuscript. A.R. thanks Wellcome Trust – DBT India Alliance for Early Career fellowship (Grant number: IA/E/18/1/504308). J.N. thanks DST-SERB for funding under Ramanujan Faculty Fellowship (Grant number: SB/S2/RJN-187/2017). A.S. also thanks IISc-Bangalore and the Ministry of Human Resource Development of India for the startup grant and the Department of Science and Technology of India for the early career grant (Grant number: ECR/2016/001702). This research was also supported by the Department of Biotechnology, Government of India in the form of IISc-DBT partnership program. Support from FIST program sponsored by the Department of Science and Technology and UGC, Centre for Advanced Studies and Ministry of Human Resource Development, India is gratefully acknowledged by the authors.

## Author contributions

A.R. and A.S. conceived and designed the research; A.R. performed research; J.N. provided computational resources; A.R., J.N., and A.S. analyzed data; and A.R. and A.S. wrote the paper.

## Competing interests

The authors declare no competing interests.
