## [Peer Review File · Nature Communications]

Reviewer #1 (Remarks to the Author):

This is fantastic work that I'm extremely excited about and looking forward to testing it. It is an excellent idea, and it is clear that the round trip times are substantially shortened in this tempering scheme relative to REST2, and thus that the REHT method has tremendous potential. I would love to see additional analyses that might explain why the base temperature replicas of Trp-cage and HIS-5 appear to be so different with REHT and REST2, as in addition to providing more confidence for early adopters of the REHT method, this might also reveal something very interesting about the nature of entropic barriers in sampling protein folding and IDPs.

It is perplexing that REHT and REST2 are not resulting in the same distributions in the base replica when using the same force field, and examining the free energy surfaces provided, it is not clear if this is just the result of increased sampling in REHT finding additional minima relative to REST2, or REST2 being stuck in different minima. It seems as though the free energy surfaces in the base replica of REHT simulations may be somewhat "flattened". This raises some concern that the tempering scheme here may not produce exactly the same equilibrium ensemble at the base temperature. Perhaps the water is not completely relaxed in the base replica using the given exchange frequency, and therefore a different distribution, where water is at a slightly different temperature, or is in different meta-stable states, is being sampled?

I would be curious to see the following additional information in the final manuscript:

- 1) A comparison of the temperature, kinetic energy and potential energy distributions of water in the base replicas of Trp-cage and HIS-5 for REST2 and REHT.
- 2) A comparison of the hydration shell properties in base replicas of Trp-cage and HIS-5 for REST2 and REHT.

In Figure S6, the free energy distributions of the base replica for Trp Cage folding are substantially different with simulations run using the same Force Field. The results do not show that REHT recovers the same distribution as REST2 with faster convergence, but that the shape of native basin is substantially different: it is broader and flatter and the free energy barriers are very different:

The author's explain this with the following "The predicted free energy barrier by the REHT method (~ 2 Kcal/mol) matches closely with the suggested free energy barrier of ~ 2.1 Kcal/mol,^{30,34} while a larger barrier of about ~ 6 Kcal/mol is observed in REST2 simulations. ", which cites an experimental study and a computational study using a different force field. Regardless of agreement to experiment or other studies with different force fields, unless the conformational space sampled is substantially different in the case of the REHT AND REST2 simulations (which does not appear to be the case), and one or both are not converged, the authors should recover the same distributions with both methods. Additionally, the figure of 6Kcal/mol for REST2 appears to be extracted from a 2D surface of RMSD vs. Rg, whereas the 1D distribution of RMSD produces a barrier of ~ 2 Kcal/mol for REST2 and ~ 1 Kcal/mol for REHT. 2Kcal/mol is in better agreement with the barrier from a 1-D contact based reaction coordinate from the cited 2011 shaw study which uses the a99SB*-ILDN force field, which is quite similar to a14SB.

Also perplexing is the dramatic difference between the Rg distributions obtained from REHT and REST2 for His-5 using the same force field. Again, unless the conformational space sampled is substantially different in the case of the REHT and REST2 simulations, this implies that one or both of

the simulations are not well converged, in which case the comparison to experimental data for unconverged simulations is somewhat irrelevant, as simulations run with the same force field with sufficient sampling should converge to the same distributions. In the final manuscript, I would be very curious to see comparisons of 2D free energy surfaces (just contours as these are easier to compare than the 3-D plots presented in the main text) of R_g vs. SASA and R_g vs. Helical content for the base replica of the REHT and REST2 simulations. This should help make it clear how similar the regions of the conformational space being sampled are, and if indeed REST2 is simply stuck in local minima of phase space, or if the two methods are somehow resulting in different distributions, different rates of discovery for different types of states in the base temperatures.

Is it possible that on the time-scales used in this study, based on the nature of the exchange schemes and higher temperature water in REHT, that REHT more readily samples extended states of IDPs in the base temperature replica, while REST2 more readily samples compact states at lower temperatures? Given enough time, we should recover the same distribution, but a bias in the types of states explored in early simulations times would in-and-of itself be an interesting result. and it would also be very interesting to attempt to understand these differences.

Paul Robustelli
Dartmouth College

Reviewer #3 (Remarks to the Author):

This is an interesting and important manuscript that reports the development and deployment of a new method REHD for improved sampling without requiring an exponential growth in the number of replicas in replica exchange MD simulations. The paper is very well written. The results are compelling and the contents are likely to be of general interest to a broad audience. The manuscript deserves to be published after some essential revisions. Please see below.

1) In the introduction, the phrase about the 90-residue long SH4 domain does not parse. Please reword.

2) It is worth noting that from a conceptual standpoint, the REHT method is akin to adding heterogeneity and hence additional degrees of freedom. This enables the mixing process and is hence akin to adding an entropy of mixing term. This analogy might be helpful.

3) It would be useful to add citations to the work of the Head-Gordon group. Please see: <https://aip.scitation.org/doi/10.1063/1.5078615>.

4) Please furnish the sequence details of Histatin 5. Also, please provide citations to work indicating that Histatin-5 adopts alpha helical structure in complex with a membrane.

5) Please specify which implicit solvent model failed in Ref. 49 so one may understand whether the indictment applies to all implicit solvation models or the one used in Ref. 49.

6) Please note that the pronouncements in Ref. 36 are badly misleading. The authors argue that water is a generic good solvent for all IDPs. This goes against a lot of data as well as simulations and is a reflection of clear issues with the coarse-grained model used by the authors. It would help if the

current authors were to refrain from advancing the trope that all IDPs are essentially extended. Please see: <http://www.jbc.org/content/early/2015/02/13/jbc.M114.619874.abstract> for suitable nuances and complexities. Also, the authors appear to convey that to be expanded is the same as being disordered. This does not have to follow because a rod-like conformation is essentially ordered if the conformational heterogeneity is low. For the IDP and RAH ensembles it would be useful to compute conformational heterogeneity measures as has been established in the IDP field c.f., <http://dx.doi.org/10.1063/1.4812791>. Also, not all IDPs will have similar scattering profiles. Please correct this misimpression created by the choice of verbiage.

7) In the discussion, why is "variety of proteins" in quotes?

8) In equation (1) X_i should not have the subscript i .

Response to reviewers

Comments from the reviewers are marked in blue italics while the authors' responses are in normal text. The content that is added to the revised text is shown in normal underlined text.

Response to comments/suggestions from Reviewer #1

Comment #1: This is fantastic work that I'm extremely excited about and looking forward to testing it. It is an excellent idea, and it is clear that the round trip times are substantially shortened in this tempering scheme relative to REST2, and thus that the REHT method has tremendous potential.

Thanks.

Comment #2: I would love to see additional analyses that might explain why the base temperature replicas of Trp-cage and HIS-5 appear to be so different with REHT and REST2, as in addition to providing more confidence for early adopters of the REHT method, this might also reveal something very interesting about the nature of entropic barriers in sampling protein folding and IDPs. It is perplexing that REHT and REST2 are not resulting in the same distributions in the base replica when using the same force field, and examining the free energy surfaces provided, it is not clear if this is just the result of increased sampling in REHT finding additional minima relative to REST2, or REST2 being stuck in different minima. It seems as though the free energy surfaces in the base replica of REHT simulations may be somewhat "flattened". This raises some concern that the tempering scheme here may not produce exactly the same equilibrium ensemble at the base temperature. Perhaps the water is not completely relaxed in the base replica using the given exchange frequency, and therefore a different distribution, where water is at a slightly different temperature, or is in different meta-stable states, is being sampled? I would be curious to see the following additional information in the final manuscript:

- i) A comparison of the temperature, kinetic energy and potential energy distributions of water in the base replicas of Trp-cage and HIS- 5 for REST2 and REHT*
- ii) A comparison of the hydration shell properties in base replicas of Trp-cage and HIS-5 for REST2 and REHT.*

We agree with the reviewer that digging deeper into the origins of shortened round trips in the new method would shed more light on why the method works better and also provide information about the nature of entropic barrier in sampling protein folding and IDPs conformation landscape. Towards that end, we are able to bring out several new facets of the method in the revised version of the work.

(i) As indicated by the reviewer, the difference in the base replicas ensemble distribution for the two methods could be due to slightly different temperature of different metastable states of the water molecules. We analyzed the distributions of potential energy, kinetic energy and temperature of water in the base replicas of both methods. The results presented in Figure R1 below show the distributions for Trp-cage and Histatin-5. The data show that the base replicas incorporate no bias in both the methods either directly or by the influence of differential tempering schemes in the higher order replicas and therefore provides similar environments for the sampling at the base replica.

Figure R1: Comparison of potential energy, kinetic energy and Temperature of water in the base replicas of Trp-cage (upper panel) and His-5 simulations (lower panel) with REST2 (red) and REHT (blue) methods.

(ii) On the other hand, the comparison of hydration shell properties of the base replicas for the two methods shed some very interesting and revealing insights into the possible origins of substantially shortened round trip times for REHT as compared to REST2. It should be noted that the trajectory at the base replica is not continuous with respect to time (the system coordinates are exchanged across different replicas at every successful exchange). Therefore, comparing the hydration shell dynamics of base replicas would give rise to incomprehensible results. Hence, we chose to do the

same analysis on the time-continuous trajectories for all replicas. We extracted and de-mixed the trajectories (1 ns each) from various time points of the simulations (For example: $t=50$ ns, 100 ns, 150 ns and up to 1000 ns). We performed the calculation of water reorientation dynamics in the hydration-shell for all the trajectories using the MDAnalysis tool. In the Figure R2 below, we show a representative set of curves on the water reorientation decay in one of the replicas (replica 0 at various time points) of Trp-cage simulated with REHT method.

Figure R2: Representative curves of water orientation relaxation at various time points of Trp-cage simulation with REHT method. The curves were fitted to biexponential decay ($C_t = A \exp^{-t/\tau_1} + B \exp^{-t/\tau_2}$) and the fitted curves are shown for each set.

The curves can be represented by a biexponential decay ($C_t = A \exp^{-t/\tau_1} + B \exp^{-t/\tau_2}$) with a set of fast (τ_1) and slow (τ_2) decay constants. We extracted these relaxation decay constants for all the sampled time-continuous trajectories of TRP-cage and His-5 with REHT and REST2 simulations and plotted their cumulative distribution in Figure R3. The distribution of fast component τ_1 (about 1 ps), which is considered relatively independent of temperature¹, exhibit similar trends in both REST2 and REHT simulations for both Trp-cage and His-5. On the contrary, the slow component τ_2 is thermally activated and its distribution is noticeably different for REST2 and REHT methods due to the differential water tempering. The activation energy for the slow relaxation (τ_2) is reduced in REHT due to differential treatment of solvent. This yields a

comparatively faster relaxation than REST2 and in some sense overcomes the “cold solvent” bottlenecks that is characteristic of REST2.

Figure R3: Cumulative distribution of fast (τ_1 in left panel) and slow (τ_2 in right panel) components of water orientational relaxation decay in REST2 (red) and REHT (blue) simulations of Trp-cage (top panel) and His-5 (bottom panel).

In general, the water relaxation is tightly coupled with the protein conformations. The orientation relaxation decay of water is slower in ordered conformations and faster in extended disordered conformations of protein. We show this coupling by plotting the protein conformations colored with the τ_2 value of water orientation relaxation in Figure R4. The water relaxation, in both the REST2 and REHT simulations, is similarly restricted in compact and ordered states. However, interestingly the major difference originates at the landscape where the entropic barrier dominates, which corresponds to the cross over regime between the intermediate state and the unfolded/disordered states (indicated by braces). This region shows relatively faster relaxation in REHT than REST2 thereby readily allowing the conformational transitions from unfolded to intermediate state. We believe that this would result in a reduced entropic barrier between the unfolded and intermediate states in REHT than REST2. As one goes further down the hill towards the compact state, the water relaxes much slower and enthalpic interactions, which tends to become

stronger with the formation of inter residue contacts (in case of folded proteins), generally drives the process of folding. The IDPs where the enthalpic compensation is weaker (due to low hydrophobicity and high net-charge), favors disordered states.

Figure R4: Tight coupling between the τ_2 decay constant and the conformational state of the protein. The conformational state of Trp-cage is indicated by the RMSD to the native NMR structure and radius of gyration (Rg). His-5 conformational state is represented along SASA and Rg collective variables. The conformations are color coded by the τ_2 value as indicated in the color bar. The region with significant entropic barrier is highlighted in all the systems. Since there are less numbers of replicas in REST in comparison to REHT (8 vs 12 in Trp-cage, 10 vs 15 in histatin-5), the number of points analyzed also became lesser in REST.

The efficiency of REST2 in crossing entropic barrier manifests much more clearly in metamorphic proteins and larger IDPs. For example, with larger IDPs (> 100 residues long as in TREK-1 C-terminal domain, unpublished data), we see that the REST2 results in eddies formed at the higher and lower zones of the replicas. Essentially exchanges between replicas take place in isolated zones – and interzonal exchanges do not take place readily. This affects the overall sampling and round trip time drastically. This behavior is also seen in the metamorphic protein Rfa-H and we have presented the exchange frequency data for the same in the paper. These results are included in the revised manuscript in section 2.4 of Manuscript.

Comment #3 (a): In Figure S6, the free energy distributions of the base replica for Trp Cage folding are substantially different with simulations run using the same Force Field. The results do not show that REHT recovers the same distribution as REST2 with faster convergence, but that the shape of native basin is substantially different: it is broader and flatter and the free energy barriers are very different:

The authors' explain this with the following "The predicted free energy barrier by the REHT method (~2 Kcal/mol) matches closely with the suggested free energy barrier of ~ 2.1 Kcal/mol (30,34) while a larger barrier of about ~ 6 Kcal/mol is observed in REST2 simulations.", which cites an experimental study and a computational study using a different force field. Regardless of agreement to experiment or other studies with different force fields, unless the conformational space sampled is substantially different in the case of the REHT and REST2 simulations (which does not appear to be the case), and one or both are not converged, the authors should recover the same distributions with both methods.

It is true that the recovered conformational landscape by REST2 and REHT methods are substantially different with respect to the shape of native basin and the free energy barrier. One of the reasons for this is REHT is less stuck at the intermediate states (due to faster water relaxation, Figure R4) and thus the sampling of native state is much faster when compared to REST2. Given longer simulation time, REST2 should recover the same energy landscape as that of REHT. So should an extremely long canonical MD simulation.

We assessed the convergence of REST2 and REHT at the physiologically relevant base replicas (@Temp 300K) similar to that suggested by Dave Thirumalai's group² and Bruce Berne's group.³ Towards this, the ensemble at the base replica is split into two equal parts, one at the start (part A) and other at the end (part B) of the total simulation (leaving a gap of 50ns between them). The two parts of the simulation represents the two independent trajectories started from different conformations of protein. The ergodicity is then measured by comparing the two simulations as follows: The conformational landscape represented by the two CVs is discretized into $m \times n$ uniform bins. The population in each of the $(i,j)^{\text{th}}$ bin ($P_{i,j}$) is compared between the two parts of the simulations A and B and the overall difference is measured using χ^2 parameter defined as: $\chi^2 = \sqrt{\sum_{i=1; j=1}^{m,n} (P_{i,j}^A - P_{i,j}^B)^2}$. If the sampling method is ergodic, the χ^2 should decay to zero. Figure R5 depicts the χ^2 as the increasing length of simulation time. The figure indicates that the REHT converges faster than the REST2 in both Trp-cage and His-5 simulations. The difference in convergence between the two methods is more accentuated in His-5 simulation.

Figure R5: Convergence of sampling in physiologically relevant base replica of Trp-cage (a) and His-5 simulations (b) with REST2 (red) and REHT (blue) methods. The CPU time including the contributions from all replicas is provided in additional X-axes.

A more rigorous measure of convergence in replica exchange simulations is to check the convergence of distributions sampled in the independent time continuous replicas. This has been shown earlier with alanine-di-peptide system,⁴ where the energy landscape is much smoother. This analysis is extremely challenging on very rugged realistic protein landscapes and demanding from sampling point of view. We performed these analyses on the time-continuous replicas of His-5 (Figure R6) and Trp-Cage (Figure R7). We plotted the distributions of R_g across the time-continuous replicas for His-5 simulations (Figure R7), which shows decent overlaps for REHT as compared to REST2. In case of Trp-cage (Figure R7), the distributions of RMSD to the native structure reveal similar trends for replicas that did not explore the folded state, whereas the replicas that fold sample distributions that are different than each other. This is observed for both REST2 and REHT methods. We anticipate that because of the complex rugged nature of the energy landscape and the existence of multiple pathways to folding. Achieving convergence of protein folding by sampling all the kinetic pathways in a single replica is a challenging task with the current level of simulation time in replica exchange simulations.⁵ We would also like to emphasize that in fact the current state-of-the-art REX methods such as REST2 and gREST explored independent folding in one or two replicas leaving most of the replicas predominantly unfolded.⁶ In that respect, the REHT definitely does a far better job by exploring the folding independently in six different replicas with the most up to date all-atom forcefields.

Figure R6: Distributions of radius of gyration at different time continuous replicas of a) REST2 and b) REHT simulations.

Figure R7: Conformational distributions of Trp-cage, plotted with the RMSD to native structure as collective variable in the time continuous replicas of REST2 (a and b) and REHT (c and d) ensembles. The distributions for the replicas that have not explored the native folded conformations in REST2 and REHT are shown in a) and c) respectively. Whereas the distributions for the replicas that explored the folding are shown in b) and d) for the REST2 and REHT simulations.

Notably, the converged sampling of a flatter landscape in intrinsically disordered His-5 is easily accessible with the REHT method compared to REST2. Extrapolation of the graphs indicated that to achieve the similar level of convergence ($\chi^2 = 0.02$), REST2 would require about 2600 ns. This would require about 12 folds more CPU times than for the REHT which attains this convergence within 150 ns. While the REHT shows similar distributions of R_g across all the time-continuous replicas, the REST2 has sampled different distributions in different replicas. Some of the replicas in REST2 is stuck at the compact states ($R_g \sim 1\text{nm}$) resulting in poor exploration of extended states ($R_g > 1.3\text{nm}$). This result provides evidence that the REST2 is possibly stuck at the entropic barrier resulting in larger variation in the R_g distributions across the replicas. We carried out further analysis on this point of view and we shall discuss this at a later stage while addressing the comments with respect of conformational heterogeneity sampling by the two method (please see response to Comment #6b By Reviewer #3).

We incorporated the convergence analysis in the revised manuscript under section S1 of supporting information and have mentioned the same in main text in page number-5 of manuscript as follows: “Also, the generated free energy maps would be meaningful only if the simulations are ergodic. We assessed the ergodicity by comparing conformational distributions of the base replica in two equal halves of the trajectory, similar to that suggested by Dave Thirumalai’s group²³ and Bruce Berne’s group¹⁰. The results suggested that the REHT converges faster than the REST2. (Please see Figure S6, Figure S7 and Section S1 in Supporting information)”

Also, the convergence of His-5 is discussed in Page number 8 of the manuscript as follows: “Moreover, to attain a converged sampling distribution, the REST2 would require 12 folds more CPU time than REHT (Figure S14-S15 and section S3).”

Comment #3 (b): Additionally, the figure of 6Kcal/mol for REST2 appears to be extracted from a 2D surface of RMSD vs. R_g , whereas the 1D distribution of RMSD produces a barrier of ~2Kcal/mol for REST2 and ~1Kcal/mol for REHT. 2Kcal/mol is in better agreement with the barrier from a 1-D contact based reaction coordinate from the cited 2011 Shaw study which uses the a99SB-ILDN force field, which is quite similar to a14SB.*

Estimating the free energy barrier is a difficult problem both experimentally as well as computationally. Finding the slowest collective variable (reaction coordinate) is a major challenge and in fact the key to solving the problem of estimating free energy barriers with high accuracy. In retrospect, we believe that it would be tenuous to make strong claims on barrier heights without having a thorough multiple-pass kinetics data. In our calculations, the reduction of free energy barrier calculated from 1D RMSD data with respect to that calculated from 2 reaction coordinates (R_g vs RMSD) indicates that the RMSD alone cannot capture the slowest reaction pathways. We qualify our earlier statements in the revised manuscript in page #5 as follows:

“However, it should be noted that the estimated barrier depends on the choice of reaction coordinates used for projecting the landscape. For the ideal reaction coordinates that capture the slowest reaction pathways, one may need to optimize the CVs with methods such as path-based sampling, and other linear (TICA, PCA) and non-linear combination of methods.”⁷⁻¹⁰

Comment #4: Also, perplexing is the dramatic difference between the Rg distributions obtained from REHT and REST2 for His-5 using the same force field. Again, unless the conformational space sampled is substantially different in the case of the REHT and REST2 simulations, this implies that one or both of the simulations are not well converged, in which case the comparison to experimental data for unconverged simulations is somewhat irrelevant, as simulations run with the same force field with sufficient sampling should converge to the same distributions. In the final manuscript, I would be very curious to see comparisons of 2D free energy surfaces (just contours as these are easier to compare than the 3-D plots presented in the main text) of Rg vs. SASA and Rg. Vs Helical content for the base replica of the REHT and REST2 simulations. This should help make it clear how similar the regions of the conformational space being sampled are, and if indeed REST2 is simply stuck in local minima of phase space, or if the two methods are somehow resulting in different distributions, different rates of discovery for different types of states in the base temperatures.

In Figure R5-R7 (under the response to comment 3a of reviewer-1), we showed the His-5 simulation with REST2 is not well converged. Whereas, the REHT recovers a converged distribution for the His-5 that can be reliably compared with the experimental averages.

The comparisons of the 2D free energy landscapes of His-5 for the base replicas of REHT and REST2 is given below (Figure R8) along the Rg versus SASA and Rg versus End-to-End distances, respectively. Since the helical content in His-5 simulation is very scarce in the simulations (Fig S13 in our supporting information document), we were not able to plot the energy landscape using this collective variable. The plots below show that the His-5 with the REST2 simulation is stuck at the compact state of the phase space whereas it explored a range of conformations in REHT methods. The 2D energy landscapes are included as Figure S12 in the supporting information of the revised version.

Figure R8: Comparison of 2d free energy surfaces of Histatin-5 ensemble constructed from the base replicas of REHT and REST2 methods shown with different collective variables such as Rg vs end to end distance and Rg with Solvent accessible surface area.

Comment #5: Is it possible that on the time-scales used in this study, based on the nature of the exchange schemes and higher temperature water in REHT, that REHT more readily samples extended states of IDPs in the base temperature replica, while REST2 more readily samples compact states at lower temperatures? Given enough time, we should recover the same distribution, but a bias in the types of states explored in early simulations times would in-and-of itself be an interesting result. and it would also be very interesting to attempt to understand these differences.

This seems to be the case. The higher temperatures of water in non-base replicas not only allow for the sampling of extended states but a range of heterogeneous conformations with frequent transitions among them. On the contrary, the REST2 seems to be trapped in an entropically

unfavorable compact state. We have tried to bring this out more clearly by plotting the time evolutions of R_g in the time-continuous replicas of His-5 simulated with REST2 and REHT methods (please see Figure R12 below while responding to Comment #6: (b) from Reviewer #3). Also, we agree that it is possible to recover the same distribution in REST2, but it will take significantly longer time than the REHT.

Taken together, we show that the difference in distribution between REST2 and REHT is (i) not because the ensemble condition at the base replica is different, (ii) but due to the faster relaxation of water in REHT that reduces the entropic barrier for conformational transitions and, (iii) also due to the lack of/slower convergence in REST2. Our method has its origin from REST2 where we try to solve the issues arising out of “cold solvent” while keeping the computational requirements tractable. REHT shows remarkable improvements in sampling converged and experimentally consistent ensembles with using the same forcefield for a variety of folding landscapes.

In this respect the following new figures has been added in the revised manuscript: Figure 6 in Main text, and Figures S6-S7, S14-S15, S22-S24.

Response to comments/suggestions from Reviewer #2

Reviewer's summary: This is a very interesting and a competently designed paper on the extension of the Replica Exchange with Solute Tempering method (REST) which will be of interest to a wide number of investigators. The original method has been used extensively to treat ligand binding to proteins with great success especially in cases where the usual parallel tempering methods do not even approximately work. It has been combined with free energy perturbation methods also with great success. In these applications the replicas of the protein are scaled but the solvent is not scaled and the overall temperature of the replicas is kept constant. This greatly reduces the number of replicas needed and thus reduces the CPU time of the simulation. For many cases this approach is fine. Yet there are many cases where the hydrogen bonding of water molecules near the protein undergoes restructuring and there it will be useful to temper the water molecules as well. This is clearly the case in the beta-hairpin where the folding and unfolding of the peptide involves the making and breaking of hydrogen bonds of water. The authors have extended the approach to accomplish this to good effect. It would be more efficient to temper only those water molecules involved near the protein but because water molecules can diffuse during the simulation this is not straight forward. A simpler approach, the one used by the authors, is to scale the overall temperature of the system in addition to scaling the Hamiltonian of the protein. This would be costly in cpu time if it weren't for the fact that one does not have to cover a wide range of temperatures.

Thanks.

Comment #1: It would be useful to compare cpu times for REHT and REST in cases where both successfully sample the conformations. This would not require much of a change. On the whole this is an outstanding paper and deserves to be published in Nature Communications.

Thanks. Estimation of the relative computational cost should be performed in fully converged simulations. However, the analysis of convergence, as discussed above as response to Comment #3a by Reviewer #1, indicates that the REST2 is not well converged as that of REHT in the timescale simulated for both Trp-cage and His-5 simulations.

We are uncertain how the trend flares out upon extending the REST2 simulation for estimating the relative computational cost. However, extrapolation of the existing data, which assumes that the trends will continue, suggests the REST2 would take about 1000 ns/replica in order for attaining the convergence level of 0.1 in TRP-cage (Figure R9). On the contrary REHT achieves this in < 500 ns. In terms of computational cost (CPU units) for TrpCage, there is only 1.3 folds reduction in REHT in comparison to REST2 considering all the replicas so for landscapes such as TrpCage, REHT and REST2 have similar overall cost. However, the real advantage is seen when proteins go through entropic barriers as in IDPs. In case of the IDP, His-5, REHT is about 12 folds faster

than REST2 (2600 ns in REST2 vs 150ns in REHT for achieving a χ^2 value of 0.02 and considering a total of 10 vs 15 replicas used). The above information has been included in supporting information under section S3 of revised version. And for metamorphic protein such as Rfa-H, the difference in REHT and REST2 is also stark, where REST2 never converges in the time we simulated.

Figure R9: Estimating relative computational cost from the convergence decay for a) Trp-cage and b) His-5 with REST2 (red) and REHT (blue). The best fit lines for each of the curves have been plotted (green and magenta) and their respective functions and parameters are indicated. The CPU time including the contributions from all replicas is provided in additional X-axes.

Response to comments/suggestions from Reviewer #3

Reviewer's summary: This is an interesting and important manuscript that reports the development and deployment of a new method REHD for improved sampling without requiring an exponential growth in the number of replicas in replica exchange MD simulations. The paper is very well written. The results are compelling and the contents are likely to be of general interest to a broad audience. The manuscript deserves to be published after some essential revisions. Please see below.

Thanks. We provide point wise response to all comments below.

Comment #1: In the introduction, the phrase about the 90-residue long SH4 domain does not parse. Please reword.

The phrase is reworded, and the revised excerpt in page number 3 reads as: “Such a powerful design of the Hamiltonian has been successfully shown to simulate weak binding of A β peptide on a lipid bilayer,¹² lateral equilibration of lipids in a bilayer,¹³, and for the generation of conformational ensemble in SH4 unique domain protein.¹⁴”

Comment #2: It is worth noting that from a conceptual standpoint, the REHT method is akin to adding heterogeneity and hence additional degrees of freedom. This enables the mixing process and is hence akin to adding an entropy of mixing term. This analogy might be helpful.

Thanks. Certainly, this analogy is useful. Further, the quantification of heterogeneity suggests that REHT precisely adds heterogeneity to the sampling and hence additional degrees of freedom (please see Figure R11 as a response to Comment #6 below). We have used this analogy in our manuscript in page number 7-8 along with the description for heterogeneity quantification.

Comment #3: It would be useful to add citations to the work of the Head-Gordon group. Please see: <https://aip.scitation.org/doi/10.1063/1.5078615>

Thanks for pointing to this relevant literature. We have cited this literature in the following context in page number 6 of manuscript. “Also, rare-event sampling methods are frequently used in conjunction with these improved force-fields to faithfully capture the conformational landscape of IDPs as shown for P53, alpha synuclein, islet amyloid polypeptide, amyloid beta, and NCBD IDPs”.

Comment #4: Please furnish the sequence details of Histatin 5. Also, please provide citations to work indicating that Histatin-5 adopts alpha helical structure in complex with a membrane.

The sequence of Histatin-5 is DSHAKRHHGYKRRKFHEKHHSHRGY. We have included this under the subsection titled “List of systems studied and preparation of atomistic model” in the main text.

Histatin-5 has not been studied with biological membrane but has been studied extensively in non-aqueous solutions such as dimethyl sulfoxide, methanol, trifluoroethanol and in model lipid vesicles such as dimyristoylphosphatidylcholine vesicles. These studies indicate that the peptide adopts α -helical structure. We apologize for the misimpression. The excerpt that read as “We also traced for the residual alpha helical propensity as observed when the peptide approaches the biological membrane during its candidacidal functioning” is now been revised as: “We also traced for the residual alpha helical propensity as observed in the non-aqueous solutions and model lipid vesicles that couples to its candidacidal functioning”

We had already cited a literature supporting the formation of helix in non-aqueous medium (Ref No: 45, Brewer et al). In the revised version, two more citations as listed below have been added.

Raj PA, Edgerton M, Levine MJ. 1990. Salivary histatin 5: dependence of sequence, chain length, and helical conformation for candidacidal activity. J. Biol. Chem. 265:3898–3905

Raj PA, Soni SD, Levine MJ. 1994. Membrane-induced helical conformation of an active candidacidal fragment of salivary histatins. J. Biol. Chem. 269:9610–9619.

Comment #5: Please specify which implicit solvent model failed in Ref. 49 so one may understand whether the indictment applies to all implicit solvation models or the one used in Ref. 49.

We now specify the name of implicit solvation model in the revised manuscript. The modified text now reads as: “the previous unbiased MD simulation, with implicit solvation model using the effective energy function EEF1¹¹, fails to produce the beta barrel spontaneously.”

Comment #6: (a) Please note that the pronouncements in Ref. 36 are badly misleading. The authors argue that water is a generic good solvent for all IDPs. This goes against a lot of data as well as simulations and is reflection of clear issues with the coarse-grained model used by the authors. It would help if the current authors were to refrain from advancing the trope that all IDPs are essentially extended.

Please see: <http://www.jbc.org/content/early/2015/02/13/jbc.M114.619874.abstract> for suitable nuances and complexities. Also, the authors appear to convey that to be expanded is the same

as being disordered. This does not have to follow because a rod-like conformation is essentially ordered if the conformational heterogeneity is low.

(b) For the IDP and RFA-H ensembles it would be useful to compute conformational heterogeneity measures as has been established in the IDP field c.f., <http://dx.doi.org/10.1063/1.4812791>.

(a) We would respectfully like to submit that we **do not** believe “*all IDPs are essentially extended*”. Also, we have not made any statement in our manuscript that proliferates the notion that “*water is a generic good solvent for all IDPs*” or convey that “*to be extended is same as being disordered*”. We regret the confusion and are not sure why the reviewer got that impression.

(b) As suggested by the reviewer, we have quantified and compared the conformational heterogeneity in His-5 and Trp-cage simulations with REST2 and REHT simulations. This has indeed provided deeper insights into the conformational landscapes. For RFA-H ensemble generated with REHT, the heterogeneity calculations have not been performed, since the comparing method REST2 fails due to inadequate sampling despite extended simulations .

To quantify the heterogeneity as suggested by Lyle et.al,¹² each of the conformation in an ensemble is represented as a vector of inter-atomic distances of all C-alpha atoms (V). The distance, D , between two conformational vectors (V_k and V_l) is then computed with the cosine distance defined as $D_{kl} = 1 - V_k \cdot V_l / |V_k| |V_l|$. The larger D value indicates more conformational heterogeneity and vice versa. For an ensemble of N conformations, the pairwise distance calculations yield $N(N-1)/2$ distance values. The distribution of D calculated for His-5 ensemble with REST2 and REHT is plotted in Figure R10. As a control, we also generated the Flory Random Coil (FRC) ensemble for the specified sequence of His-5 using Campari tool and plotted their distance distribution. 5000 conformations were generated with $3 \cdot 10^7$ Monte Carlo steps after discarding first 500000 steps of equilibration. Our results indicate larger heterogeneity for the REST2 ensemble than for the REHT and FRC, which at face value is quite counterintuitive. We delve into the origin of it shortly.

Figure R10: Distributions of heterogeneity values calculated using Cosine distance (a) and pairwise q-factor (b). The heterogeneity of the ensembles generated with the base replicas of REST2 (red), REHT (blue) methods have been compared against the control FRC ensemble (black). The larger D and smaller q represent greater heterogeneity in the ensemble.

To verify this result, we also used a second heterogeneity measure described by Papoian et.al.¹³ According to this metric, the heterogeneity (q) between two conformational vector k and l is computed with the differences in pair distance values ($r_{a,b}$) as follows: $q_{kl} = \frac{1}{N_{pairs}} \sum_{a,b} \exp \left[-\frac{(r_{ab}^k - r_{ab}^l)^2}{2\sigma^2} \right]$, where the σ is a resolution parameter and usually is set to 2\AA . Consistent with the cosine distance distribution (Figure R10a), the histogram of the pairwise- q (Figure R10b) also reveal larger heterogeneity (low q) for REST2 than the REHT and FRC. However, the free energy landscape of His-5 shown with Rg and SASA (Figure R8) suggests a confined sampling of compact states in REST2. These counterintuitive observation (seemingly larger heterogeneity for REST2) and conflicting results (w.r.t the corresponding free energy landscapes) needed to be reconciled. We anticipated that one of the possible explanations for this inconsistency could have its origin in the “*heterogeneous compact structures*” that the REST2 simulations sample for the His-5 ensemble.

To inspect this, we transformed the cosine distances (D_{kl}) between all pairs of conformations ($n \cdot (n-1)/2$ values) into a 2-dimensional map of n conformations using multidimensional scaling (MDS).¹⁴ MDS is particularly useful for visualizing the distance matrix in low dimensional space while preserving the between-object distances as much as possible. The conformational map along the MDS coordinates is presented in Figure 10 for both REST2 and REHT. Coloring each conformation with the respective Rg value (Figure 10a and c) clearly shows that there are more and diverse compact states in REST2 (than REHT) that spread all over the space. While this explains the heterogeneity of REST2, it poses an additional question of how REST2, which tends to get stuck in barriers, sample heterogenous compact states? (heterogeneous compact states typically have larger energetic as well as entropic barriers). To answer this, we colored the

conformational map of REST2 and REHT by the replica index in figure 10 c and d. In REST2, different replicas occupy a non-overlapping and maximally separated space (replica number 2, 3, 4 and 7). This suggests that the different replicas simulated with REST2 are stuck at different local basins and therefore provide distinct compact conformations to the base replica upon the successful exchanges. Whereas in REHT, the contributions from independent replicas are not significantly different/distant from each other while within a single replica they sample a diverse space. Therefore, in REHT the heterogeneity arises by the exhaustive exploration of conformations in independent replicas; whereas in REST2 the enhanced heterogeneity arises in the base replica by the virtue of exchange of conformations between replicas that are stuck at different local basins of compact structures.

Figure R11: Configurational map of REST2 (a, c) and REHT (b, d) ensemble across 2-dimensional MDS axes. In a and b the conformations are colored by the respective Rg values and in c and d they are colored by replica index. Maximally separated non-overlapping clusters in REST2 are marked in c.

The conformational trap in REST2 can also be seen from the evolution of R_g in the time-continuous replicas as plotted in Figure R12. The Figure R12 shows that most of the replicas in REST2 are not sampling the dynamically heterogeneous population (unlike REHT as shown in Fig R12b) but rather persistently stuck at the compact states (very evident in replica 3 and 4 for example). This conformational trap at the compact states in REST2 accounts for its poor convergence as shown from Figure R5.

The above data has been included in Figures 4 and S16-S17 and discussed in page number 8 and 18 of main text and in section S4 of Supporting information of revised manuscript.

Figure R12: Evolution of R_g across the time-continuous replicas of His-5 simulated with REST (a) and REHT (b) methods. REST shows the trapping of compact states in multiple replicas; whereas, the REHT rapidly explores the dynamically heterogeneous conformations in all the replicas.

For the sake of completion, a similar heterogeneity analysis (using Papoian measure¹³) was also performed for Trp-cage and the results are presented in Figure R12. The analysis reveals two major peaks for the ensemble sampled with REST2, corresponding to the unfolded and folded states with a minor peak corresponding to the intermediate state. Whereas in REHT, additional peaks were observed indicating multiple intermediate states being sampled in the simulation. Further, the heterogeneity calculated in different time blocks of the base replica indicates that these heterogeneous populations of REHT appeared very early in the simulation when compared to REST2. These results further support our results from Figure R4, that the transitions from unfolded

to intermediate states of Trp-cage protein are made easy with the improved reorientation dynamics of water.

Figure R12: Comparison of pairwise q distributions for the Trp-cage ensembles obtained from the base replicas of REST and REHT methods. A control distribution for the FRC ensemble is also included. The distributions of heterogeneity measured in different blocks of the base replica from REST2 and REHT is plotted in (b) and (c) respectively.

Comment #6 (c):) Also, not all IDPs will have similar scattering profiles. Please correct this misimpression created by the choice of verbiage.

Thanks. We have corrected the text in Page number-7 to the following: “In case of compact well folded proteins, the Kratky plot exhibits a bell-shaped peak at low- q regime and converges to the q -axis at high- q regime. Conversely the disordered proteins, depending on the degree of compactness, flexibility of the chain and the presence of structured regions show different curves. For the completely expanded or fully unfolded proteins the intensity at high- q region exhibits a plateau, that may be followed by further rise in some cases.^{15,16”}

Comment #7: In the discussion, why is "variety of proteins" in quotes?

The quotes have been removed.

Comment #8: In equation (1) X_i should not have the subscript i .

The script is corrected in the equation.

References:

1. Yeh, Y. & Mou, C.-Y. Orientational Relaxation Dynamics of Liquid Water Studied by Molecular Dynamics Simulation. *J. Phys. Chem. B* **103**, 3699–3705 (1999).
2. Thirumalai, D., Mountain, R. D. & Kirkpatrick, T. R. Ergodic behavior in supercooled liquids and in glasses. *Phys. Rev. A, Gen. Phys.* **39**, 3563–3574 (1989).
3. Liu, P., Kim, B., Friesner, R. a & Berne, B. J. Replica exchange with solute tempering: a method for sampling biological systems in explicit water. *Proc. Natl. Acad. Sci. U. S. A.* **102**, 13749–13754 (2005).
4. Gil-Ley, A. & Bussi, G. Enhanced conformational sampling using replica exchange with collective-variable tempering. *J. Chem. Theory Comput.* **11**, 1077–1085 (2015).
5. Paschek, D., Nymeyer, H. & Garcia, A. E. Replica exchange simulation of reversible folding / unfolding of the Trp-cage miniprotein in explicit solvent : On the structure and possible role of internal water. **157**, 524–533 (2007).
6. Kamiya, M. & Sugita, Y. Flexible selection of the solute region in replica exchange with solute tempering: Application to protein-folding simulations. *J. Chem. Phys.* **149**, (2018).
7. Best, R. B. & Hummer, G. Reaction coordinates and rates from transition paths. *Proc. Natl. Acad. Sci. U. S. A.* **102**, 6732–6737 (2005).
8. Pérez-Hernández, G., Paul, F., Giorgino, T., De Fabritiis, G. & Noé, F. Identification of slow molecular order parameters for Markov model construction. *J. Chem. Phys.* **139**, 15102 (2013).
9. Tiwary, P. & Berne, B. J. Spectral gap optimization of order parameters for sampling complex molecular systems. *Proc. Natl. Acad. Sci. U. S. A.* **113**, 2839–2844 (2016).
10. Chen, W., Sidky, H. & Ferguson, A. L. Nonlinear discovery of slow molecular modes using state-free reversible VAMPnets. *J. Chem. Phys.* **150**, 214114 (2019).
11. Lazaridis, T. & Karplus, M. Effective energy function for proteins in solution. *Proteins Struct. Funct. Bioinforma.* **35**, 133–152 (1999).
12. Lyle, N., Das, R. K. & Pappu, R. V. A quantitative measure for protein conformational heterogeneity. *J. Chem. Phys.* **139**, 121907 (2013).
13. Potoyan, D. A. & Papoian, G. A. Regulation of the H4 tail binding and folding landscapes via Lys-16 acetylation. *Proc. Natl. Acad. Sci. U. S. A.* **109**, 17857–17862 (2012).
14. Borg, I. & Groenen, P. *Modern multidimensional scaling : theory and applications.* (Springer-Verlag, 2005).
15. Receveur-Brechot, V. & Durand, D. How Random are Intrinsically Disordered Proteins? A Small Angle Scattering Perspective. *Curr. Protein Pept. Sci.* **13**, 55–75 (2012).
16. Banks, A., Qin, S., Weiss, K. L., Stanley, C. B. & Zhou, H. X. Intrinsically Disordered Protein Exhibits Both Compaction and Expansion under Macromolecular Crowding. *Biophys. J.* **114**, 1067–1079 (2018).

Reviewer #1 (Remarks to the Author):

I am pleased by the extremely thorough and rigorous response by author's to all of my comments, as well as the comments of the other reviewers, and I enthusiastically recommend publication at this time.

Paul Robustelli
Dartmouth College

Reviewer #3 (Remarks to the Author):

The authors have addressed all of the comments / concerns I raised. The heterogeneity analysis has proven to be quite insightful. The demonstration that REST2 engenders heterogeneous compact conformational traps is really interesting. Figure 10 and the accompanying analysis in the SI will be invaluable. The use of MDS was really clever and the findings reported here will be of immense value to the simulation community and for quantifying / distinguishing heterogeneity / disorder of the glassy variety vs. heterogeneity encoded by the energy landscape. I am thinking of the distinction between distributions of barriers vs. the multiple minima problem. Overall, given the extensive revisions, and the responsiveness of the authors, I urge publication post haste.